# An Inadequate Blood Supply Is a Risk Factor of Anastomotic Biliary Strictures After Liver Transplantation—A Single-Center Study

**DOI:** 10.3390/jcm14041365

**Published:** 2025-02-18

**Authors:** Samir Zeair, Marek Mamos, Julia Hirchy-Żak, Patryk Modelewski, Robert Stasiuk, Mariola Post, Artur Uździcki, Michał Witkowski, Agata Łakomiak, Marta Wawrzynowicz-Syczewska

**Affiliations:** 1Department of General and Transplant Surgery, Pomeranian Regional Hospital, 71-455 Szczecin, Poland; samirzeair@gmail.com (S.Z.); r.stasiuk@yahoo.pl (R.S.); mariolapost@wp.pl (M.P.); micwitkowski@spwsz.szczecin.pl (M.W.); 2Department of Infectious Diseases, Hepatology and Liver Transplantation, Pomeranian Medical University, 71-455 Szczecin, Poland; marekmamos3@gmail.com (M.M.); hirchyj@gmail.com (J.H.-Ż.); model66434@o2.pl (P.M.); agata.lakomiak@wp.pl (A.Ł.); 3Department of Internal Medicine and Gastroenterology, Pomeranian Regional Hospital, 71-455 Szczecin, Poland; arturuzdzicki@gmail.com

**Keywords:** liver transplantation, biliary complications, anastomotic biliary strictures, endoscopic retrograde cholangiopancreatography

## Abstract

**Background**: Anastomotic biliary strictures (BSs) are among the most common complications after liver transplantation (LT), accounting for 5–15% of adult recipients after deceased-donor transplantation. For some reason, this percentage increased in our center in recent years, and the goal of this study was to find out the reasons behind this to avoid this complication in the future. **Material and Methods**: We retrospectively analyzed the occurrence of anastomotic biliary strictures in 230 cadaveric-donor LTs performed in our center between January 2019 and December 2023. Many variables related to the donor, recipient, and surgical procedure were compared between patients who experienced BS and those without this complication. Statistical analysis was performed using Fisher’s exact test, a one-way ANOVA test, and Pearson’s correlation coefficient. **Results**: Altogether, 51 patients (22.17%) developed BSs. This percentage was especially high in 2023 (32%). The only significant differences found in study group compared to the control group were the requirement of additional doses of vasopressors during surgery (45 (86.53%) vs. 138 (77.09%), *p* = 0.0001) and more frequent instances of reperfusion syndrome (8/51 (15.68%) vs. 11/179 (6.11%), *p* = 0.00001). **Conclusions**: We conclude that ischemia during LT has an advantage over technical parameters in the development of BSs after LT. Appropriate blood volume resuscitation as opposed to inotropic treatment may reduce the risk of this complication.

## 1. Introduction

Liver transplantation (LT) is a routine treatment of end-stage liver disease. Albeit this procedure is a real success story, several post-transplant complications, influencing long-term survival, occur. The so-called Achilles’ heel of liver transplantation with a major influence on morbidity and mortality are biliary complications. Their frequency ranges from 5 to 15% after deceased-donor transplantation depending on the type of biliary reconstruction, the etiology of liver disease, and the vascularization of bile ducts, reaching 32% after living-donor liver transplantation [1]. Post-LT biliary complications include biliary strictures (BSs), bile leakage, formation of stones, and Oddi sphincter dysfunction, with biliary strictures being the most frequent [2]. The most common biliary anastomosis is choledochocholedochostomy (CC), with or without T-tube reconstruction, followed by choledochojejunostomy (CJ) [3]. The choice of anastomosis depends on the underlying liver disease: CJ is usually performed in children with biliary atresia and in primary sclerosing cholangitis when the recipient’s bile ducts are grossly damaged. Discrepancy between donor and recipient bile duct size is another reason for CJ anastomosis.

Post-LT biliary strictures are classified as anastomotic, located at the site of biliary anastomosis, and non-anastomotic, the so-called ischemic-type strictures, hilar in location or intrahepatic. Strictures can be seen early after transplantation, but usually, they occur approximately 5–8 months after surgery [4]. CC anastomosis can be performed in two fashions: end-to-end or side-to-side [5]. The use of T-tube drainage in duct-to-duct biliary reconstruction is still being discussed by researchers, and many transplant centers have now abandoned this technique due to the high number of leakages after drain removal [6]. A recent randomized controlled trial, however, favored T-tube insertion in the side-to-side type of CC reconstruction [7]. The clinical presentation of BSs ranges from asymptomatic biochemical cholestasis to major biliary complications such as ascending cholangitis with fever, abdominal pain, jaundice, and pruritus.

The study on biliary complications performed in our center in 2014–2017 showed that the frequency of BCs, including strictures, choledocholithiasis, and biliary leakage, was comparable to that found in other European centers, accounting for 17.6% of the total [8]. In the year 2023, our transplant team was alarmed by hepatologists and endoscopists about the increasing number of endoscopic retrograde cholangiopancreatography (ERCP) procedures needing to be performed in liver recipients due to clinically symptomatic BSs. The aim of this study was to elucidate factors related to this unfavorable phenomenon.

## 2. Material and Methods

We retrospectively analyzed the outcome of cadaveric-donor LTs performed in our center between January 2019 and December 2023, specifically the occurrence of anastomotic biliary strictures (BSs) requiring endoscopic treatment. All patients gave their informed consent to participate in this study. Ethical approval was not required because of the standard procedure. All recipients received full-size liver graft, and the transplantation technique employed was predominantly the piggy-back one; only four LTs were performed classically. University of Wisconsin solution was used for liver preservation. The composition of the surgical team did not change during the time of this study. Biliary reconstruction was performed by end-to-end anastomosis or hepaticojejunostomy using a continuous 6.0 Monoplus suture. All CC anastomoses were performed without T-tube drainage. No side-to-side CC anastomoses were performed. Cases with hepatic artery thrombosis, bile leakage, re-transplantation, and early death (<30 days from LT) were excluded from the analysis. The inclusion criteria were patients with elevated cholestatic enzymes, such as alkaline phosphatase (ALP) and gamma-glutamyltransferase (GGT), and the occurrence of clinical symptoms such as pruritus, jaundice, pale stools, abdominal pain, fever, and diarrhea on some occasions. Clinical and laboratory information was obtained from the medical files. To confirm the diagnosis of BS, cholangio-magnetic resonance and/or ERPC was performed (Figure 1). All ERCP procedures were performed by two experienced endoscopists. The equipment used for ERCP was the Olympus duodenoscope model EVIS EXERA II TJF-Q190V (Tokyo, Japan). In the case of BS, balloon dilatation and stenting were performed (Figure 2 and Figure 3).

Variables of donor, recipient, and surgical procedure were analyzed. The variables related to the recipient were the following: age, sex, cholestatic or non-cholestatic type of liver disease etiology, and MELD score. The gender, age, and length in ICU stay of the donor were taken into account. The factors related to the transplant procedure were the following: cold ischemia time (CIT), second warm ischemia time (WIT), number of red blood cell (RBC) units transfused during surgery, type of hepatic artery anastomosis (classical or variant), presence of reperfusion syndrome, additional use of vasopressors (other than the routine single dose routinely given during graft reperfusion), mean arterial pressure (MAP) before reperfusion, and the lowest MAP five minutes after reperfusion. Classical hepatic artery anastomosis was performed on the patch between the common and gastroduodenal hepatic artery of the recipient and the donor. Any other arterial anastomoses were considered variants.

## 3. Statistical Analysis

All calculations were performed using the SPSS software package (IBM SPSS Statistics, 27.0 version, IBM, Sheffield, Great Britain). Categorical variables were expressed as numbers and percentages. Continuous variables were expressed as means and 95% confidence intervals or as medians with interquartile ranges (IQRs). A Shapiro–Wilk test was used to check for normal distribution. The mean or median values for the study group and the control group were compared with a one-way ANOVA test. Comparisons of nominal variables were performed with Fisher’s exact test. In this study, group correlation between CIT, WIT, and MAP before reperfusion and the time elapsed from transplantation to ERCP was performed using Pearson’s correlation coefficient. A *p*-value < 0.05 was considered statistically significant.

## 4. Results

Two hundred and thirty nine consecutive adult patients who had received a transplant in the Provincial Hospital in Szczecin, Poland, between January 2019 and December 2023 were studied retrospectively. Nine patients were excluded from the analysis (six early deaths and three re-transplantations). Finally, data from 230 patients, 135 males (58.7%) and 95 (41.3%) females, with in a median age of 58 years (IQ 46–66), were available. Characteristics of the overall group are shown in Table 1.

All patients received a full-size liver graft from donors who had been declared deceased following brain death. Patients with the diagnosis of symptomatic anastomotic biliary stricture, requiring endoscopic treatment, formed the study group. Patients who did not need ERCP served as the control group. The diagnosis of BS was based on clinical and biochemical abnormalities, confirmed by sensitive modalities such as magnetic retrograde cholangiopancreatography (MRCP) and/or direct cholangiography with ERCP.

The study group comprised 51 patients (22.17% of the total number of patients transplanted in the years 2019–2023 included in this study). The number of anastomotic BS noted each year in the study period (2019–2023) was 14 out of 59 (23.7%), 2 out of 24 (8.3%), 5 out of 36 (13.9%), 11 out of 49 (22.4%), and 19 out of 59 (32%), respectively. The median time from LT to ERCP was 151 days (IQR 51–265 days). There were 16 cases of early BS (<3 months after LT) and 35 late-presented strictures (>3 months after LT). Each patient from the study group had end-to-end biliary anastomosis without T-tube insertion; hepaticojejunostomy was not performed.

The study (51 patients) and the control (179 patients) groups were compared with respect to the age and sex of the recipients, the etiology of liver disease (cholestatic vs. non-cholestatic), donor parameters (sex, age, and length of stay in ICU), and factors related to the procedure (CIT, WIT, biliary splintage, type of hepatic artery anastomosis, mean arterial pressure before and after reperfusion, use of vasopressors, and number of RBC units transfused during surgery). These data are presented in Table 2.

In all cases but four, LT was performed using the piggy-back technique. The classical technique was performed in one patient from the study group and in three patients from the control group (NS). CJ anastomosis was performed only in three patients from the control group and in nobody from the study group. No difference in age, sex, and cholestatic vs. non-cholestatic etiology of the disease was found between the study and control groups. There was no difference between donors’ age, sex, and length of ICU stay. The use of splintage and the type of hepatic artery anastomosis were comparable between groups. The number of RBC units transfused during transplantation did not differ significantly.

In the study group, more patients required additional doses of vasopressors (45 (86.53%) vs. 138 (77.09%), *p* = 0.0001), according to Fisher’s exact test. Moreover, patients from the study group presented significantly more often with reperfusion syndrome (8/51 (15.68%) vs. 11/179 (6.11%), *p* = 0.00001). Another difference was a longer CIT in the group with ERCP, with a median time of 373.91 min (95% CI 340.36–407.46) vs. 344.64 min (95% CI 323.93–365.45), but this difference did not reach statistical significance (*p* = 0.09). The mean arterial pressure before reperfusion was comparable. However, there was statistically significant correlation between the MAP measured directly before reperfusion and the number of days that elapsed between LT and ERCP (Pearson’s coefficient of 0.3, *p* = 0.03). These data are shown in Table 3.

Some parameters, such as CIT, WIT, type of the hepatic artery anastomosis, and number of RBC units transfused during surgery, were compared between patients who had presented with early BS and late presenters. These data are shown in Table 4. No difference was identified.

## 5. Discussion

Our study confirms the increased number of biliary complications that happened in the analyzed time frame compared to a previous study from the same transplant center (15.4 vs. 22.2%). A particular increase (32% incidence) was noted in the year 2023. It needs to be emphasized that only anastomotic biliary strictures were analyzed. Other biliary complications, such as biliary leakage, choledocholithiasis, or non-anastomotic strictures, were excluded from the given study. It is also noticeable that, during the pandemic era, the number of BSs and LTs in total was rather low, probably due to the more careful selection of both donors and recipients forced by the limited access to intensive care units and transportation means.

There are several risk factors related to the development of BSs, such as the reconstruction technique, technical errors, underlying liver disease, donor characteristics, preservation of the organ, and hemodynamic conditions during surgery, among others. Anastomotic strictures (ASs) are located within one centimeter from the surgical suture, and it is important to leave enough length of the ducts to avoid tension. Another crucial thing is to guarantee an appropriate blood supply throughout the transplantation procedure. According to Verdonk et al., the occurrence of ASs in deceased-donor LT is around 4–9% [9]. In another study, ASs in end-to-end anastomosis were noted in 22% of recipients, half of them being late-onset [5]. A relatively high number of BCs, including strictures, is related to living-donor liver transplantation, ranging from 5 to 37% [10,11]. In our study, we aimed to figure out factors that might explain the increasing frequency of biliary strictures in order to avoid this complication in the future and improve outcomes.

In cases of anastomotic BSs, it is reasonable to incriminate technical issues as the main cause of the problem. There are no unequivocal guidelines for the best type of biliary reconstruction technique, and considerable differences and preferences exist among surgeons. After careful analysis of the surgical aspects of biliary anastomosis, the only difference identified in the currently analyzed group compared to the previous study was the site where the common bile duct had been harvested. Currently, it is cut slightly above the connection with the cystic duct, while previously, the bile duct was cut a few millimeters below. The donor’s bile duct is considered worse in terms of blood supply than the bile duct of the recipient, and as a preventive strategy against ischemia, it has been shortened more to ensure a valid blood supply. No other difference in the surgical technique was identified compared to the 2014 study. All patients received a whole liver from a patient who had been declared deceased following brain death. The preservation solution was the same in each case, and the type of anastomosis, method of performing sutures, and the surgical materials did not change over time. A continuous absorbable suture was preferred over an interrupted suture with non-absorbable monofilaments [12]. Patients were operated on by the same transplant team. The study and control groups did not differ in terms of age, gender, and severity of liver disease. The instances of cholestatic and non-cholestatic etiologies of liver disease were comparable between groups. There were no differences in donors parameters between the study and control groups.

All BSs in our study occurred relatively early after LT. The mean interval for developing a BS was 5 months. This is consistent with other findings showing the most common time of BS appearance being within the first year post LT [13]. Early BSs (<3 months) are mostly related to the mismatch between the donor’s and recipient’s size of bile ducts. Late BSs are explained by a poor blood supply to the anastomosis with subsequent fibrosis [14]. Patients from the study group with early and late BSs were compared in terms of the type of hepatic artery anastomosis (classical vs. variant) possibly explaining ischemia of the anastomotic site. There was no difference in the number of classical and variant hepatic artery anastomosis performed in the early and late presenters. This difference was also not found between the whole study group and the control group (39 vs. 37%, *p* = 0.83). There was no difference in the number of blood units transfused during surgery between the study and control groups or between early and late presenters of BSs (Table 4).

Because technical issues did not seem to play a major role in the development of BSs in our series, it is probable that some ischemic reasons might need to be considered. The influence of ischemia as a cause of anastomotic strictures was not as strong as in the case of non-anastomotic strictures [15], but there were some suggestions that ischemia might have contributed to the development of BSs in the analyzed group of patients. The use of additional vasopressors during surgery and more frequent instances of reperfusion syndrome in the study group indicated arterial perfusion problems that might have been responsible for the damage of bile ducts being supplied solely by the hepatic artery. One of the proposals to overcome the problem of an inadequate blood supply to the bile ducts is to resuscitate blood volume with RBC infusions instead of using vasopressors that can cause unnecessary vasoconstriction. The impact of CIT on the development of BSs in our series was inconclusive. It was found that CIT longer than 8 h is a strong predictor of ischemic cholangiopathy [16]. However, the mean CIT in our analysis was relatively short (around 6 h), and the difference in CIT between the study and control groups did not reach statistical significance. Nonetheless, the CIT in the study group was slightly longer (373.9 vs. 344.6 min). We analyzed the same problem in our previous study [8], where we obtained similar results, so we concluded that minor changes in the CIT between the studied groups (up to several minutes) did not contribute to the ischemic nature of the biliary strictures in our series.

The limitation of our study was in the relatively small number of patients, preventing us from making the statistical analysis strong, alongside the omission of the problem of infections as a cause of stenosis. It was shown that infections more often contribute to the development of NAS [17]. Moreover, the elimination of HCV recurrence as a cause liver fibrosis and bile duct damage suggest that infections are less important in the development of anastomotic BSs, leading to the exclusion of this potential risk factor seeming warranted.

The prevention of biliary strictures is a complex problem related to the perioperative management of patients. Preventive strategies are still being debated. One of the conclusions from our study is the need for better filling of the vascular bed. It is also worth considering whether bile duct shortening might be behind the increased number of biliary strictures or whether it is rather beneficial. Therefore, for future research, a comparative study with two methods of donor bile duct collection (as described above) is planned.

## Figures and Tables

**Figure 1 jcm-14-01365-f001:**
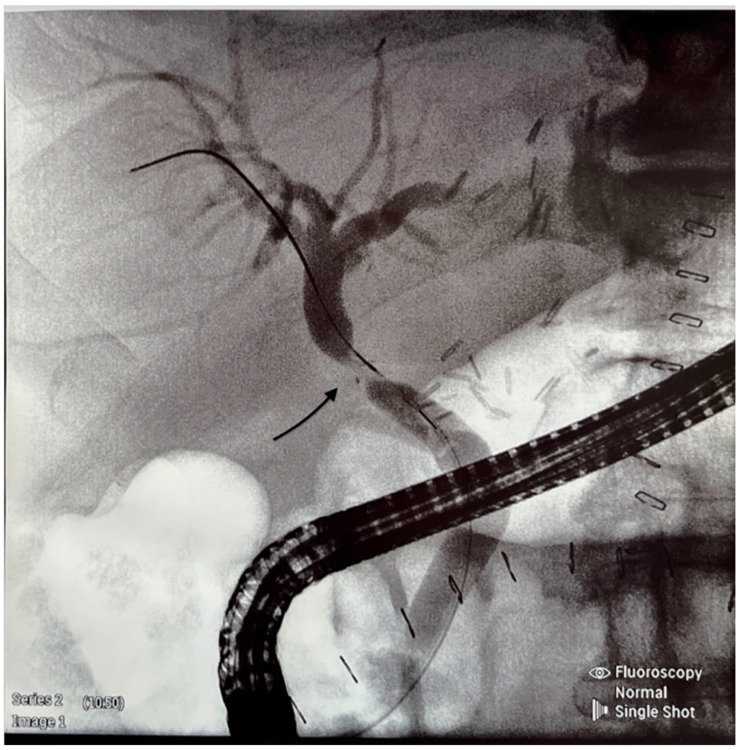
Endocopic retrograde cholangiogram shows an anastomotic stricture (see arrow).

**Figure 2 jcm-14-01365-f002:**
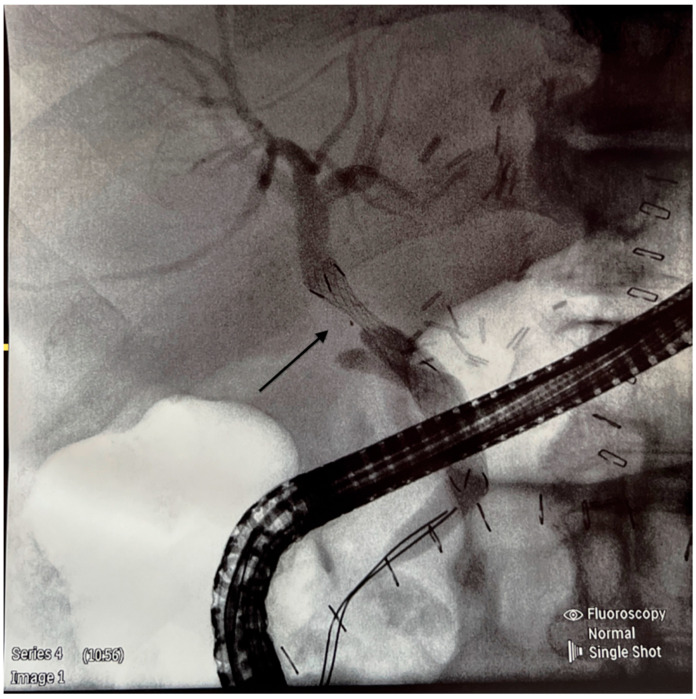
A metal stent is inserted across the stricture as shown by the arrow.

**Figure 3 jcm-14-01365-f003:**
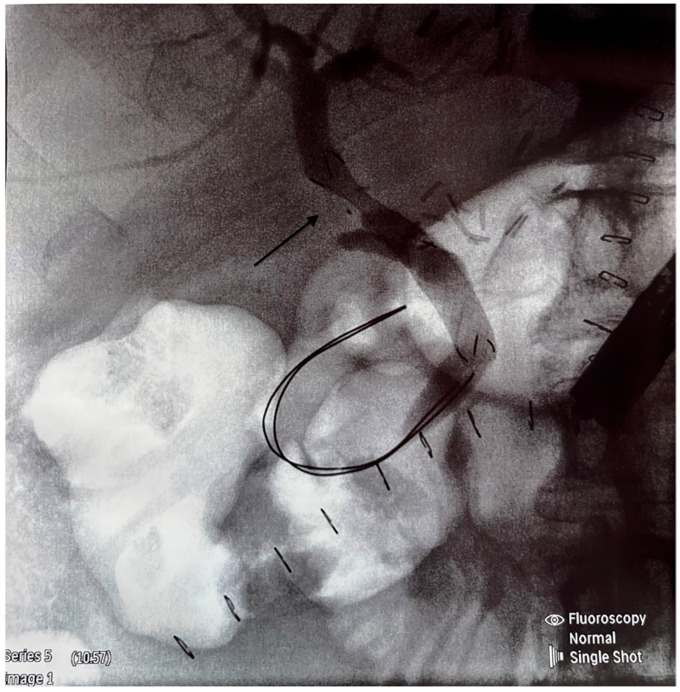
Good effect of stenting (free flow of contrast as shown by the arrow).

**Table 1 jcm-14-01365-t001:** Characteristics of the recipients, n = 230.

	Median or Numbers	%, 95% CI or IQR
Age [median, IQR]	58	46–66
Gender [male, %]	135	58.7
MELD [mean, 95% CI]	14.9	6.0–23.4
Etiology of liver disease:		
Cholestatic [n, %]	32	13.9
Non-cholestatic [n, %]	198	86.1

**Table 2 jcm-14-01365-t002:** Comparison of study parameters between study group and controls.

	Patients Requiring ERCP, n = 51	Controls, n = 179	Statistical Significance
Age in years [median, IQR)	56 (48.75–67.25)	58 (45–66)	NS
Male sex [n, %]	32 (63)	103 (58)	NS
MELD [mean, 95% CI]	13.9 (6.4–24.1)	14.5 (6.1–23.8)	NS
Age of donors in years [median, IQR]	49.5 (37.5–60.75	47 (36.5–60)	NS
Male sex of donors [n, %]	32 (63)	115 (64)	NS
Donor ICU stay in days (mean, 95% CI)	4.96 (3.68–6.24)	5.41 (4.66–6.16)	NS
CIT in min [mean, 95% CI]	373 (340.37–404.47)	344.64 (323.94–365.35)	NS
WIT in min (mean, 95% CI)	30.5 (29.14–31.86)	31.93 (30.96–32.9)	NS
Splintage of bile ducts [n, %]	11 (21,56)	48 (26.81)	NS
MAP before reperfusion in mmHg [mean, 95% CI]	70.58 (66.19–74.97)	70.99 (68.72–73.26)	NS
Additional use of vaso-pressors [n, %]	45 (86.53)	138 (77.09)	***p* = 0.0001**
Reperfusion syndrome [n, %]	8 (15.68)	11 (6.11)	***p* = 0.00001**
Acidosis correction [n, %]	35 (68.63)	133 (74.3)	NS
Classical hepatic artery anastomosis [n, %]	30 (63)	107 (61)	NS
Variant hepatic artery anastomosis [n, %]	19 (37)	63 (39)	NS
Patients requiring blood transfusion [n, %]	34 (69)	113 (66)	NS
Median number of RBC units	2	1.91	NS

**Table 3 jcm-14-01365-t003:** Correlation between time (in days) elapsed between LT and ERCP and the chosen parameters.

Study Parameter	Pearson’s Correlation Coefficient	*p*-Value
CIT in min	0.02	0.87
WIT in min	0.3	0.67
MAP before reperfusion in mmHg	−0.3	**0.03**
The lowest MAP 5 min after reperfusion in mmHg	−0.11	0.45

**Table 4 jcm-14-01365-t004:** Comparison of the chosen study parameters between patients with early and late BS.

	Early BS, n = 16	Late BS, n = 35	Statistical Significance
Donor ICU stay in days [mean, 95% CI]	4.33 (2.54–6.16)	4.97 (3.48–6.47)	NS
Median age of recipients	58.5	56	NS
Classical hepatic artery anastomosis [n, %]	11 (68.75)	21 (60)	NS
Variant hepatic artery anastomosis	5 (31.25)	14 (40)	NS
Requirement of blood trans-fusion [n, %]	9 (56.25)	24 (69)	NS
Median number of RBC	1.88	2	NS

## Data Availability

The datasets generated during and/or analyzed during the current study are not publicly available, but are available from the corresponding author on reasonable request; no additional data are available.

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
