# Peer review of "An Inadequate Blood Supply Is a Risk Factor of Anastomotic Biliary Strictures After Liver Transplantation—A Single-Center Study"

_jcm, 2025, doi:10.3390/jcm14041365_

Round 1

Reviewer 1 Report

Comments and Suggestions for Authors

The authors investigated ischemic events rather than technical aspects as risk factors for anastomotic biliary stricture after liver transplantation. I have several comments regarding this.

1. The authors investigated ischemic events rather than technical aspects as risk factors for anastomotic biliary stricture after liver transplantation. Although this study aims to investigate the reasons for the increased incidence of biliary stricture, the absolute incidence of biliary stricture seems too high. The higher biliary stricture rate makes it difficult to disregard technical factors. It would be beneficial to discuss this aspect in more detail. 

2. It would be helpful to provide more detailed information on the biliary reconstruction methods. Specifically, it would be valuable to clarify whether interrupted or continuous sutures were used, the thickness of the sutures, the degree of tension applied, and so on. Also, please describe the strategy used in donor-recipient bile duct size mismatch cases.

3. The study reports a high incidence of early biliary stricture events, and given that deceased donor liver transplantation (DDLT) generally has better vascularity in the surrounding bile duct tissue, I am not fully convinced by the authors’ conclusions only focusing on ischemic issues. However, it is clear that vascularity to the bile duct is an important issue. Please clarify the detailed meaning of "additional vasopressors" mentioned in their study and the definition of "reperfusion syndrome".

Author Response

Thank you very much for reviewing our manuscript. We agree that the reason for such high number of biliary strictures in our series is rather unresolved and following your suggestion we had a thorough analysis of the procedure with particular attention to technical aspects. A few years ago our team decided to implement a little change in the donor's bile duct harvesting cutting it a bit shorter. Rationale for this was the desire to improve blood supply to the donor's bile duct. Now the doubt arose as to whether it was advantageous. Generally speaking biliary complications seem to be all about ischemia and the whole team effort should focus on ensuring an appropriate blood supply.

Reviewer 2 Report

Comments and Suggestions for Authors

Dear authors

This is a very interesting research and I congratulate you for it.

I have a few remarks regarding your manuscript:

- put the tables 1-4 in the Results section, near the text where they are mentioned, not after Discussions; 

- bold the significant p-values from the tables (the two results from table 2 and the result from table 3);

- I believe there is an error in row 126; the study group has 51 patients and the control has 179 patients, not the way you mentioned it; 

- please check the text for minor spelling errors (e.g. ERPC in row 79);

- do you have images from the endoscopic procedures (ERCP)? It would comprise a good addition to the manuscript and it would enhance its scientific value;

- at the end of Discussions, insert a paragraph about future research directions;

- mandatory - insert a Conclusion section where you briefly present your most important findings from the study and the main ideas from the research.

Author Response

Answers to the 2 reviewer:
Thank you very much for your comments and your thorough paper evaluation. Please, find our answers:
Ad.1 Tables 1-4 were moved into the Results section next to the appropriate results presentations
Ad. 2 Significant  p values from tables 2 and 3 were bolded
Ad 3. The error in row 126 was corrected - we indeed swapped results of the study group and the control group
Ad 4. Error spelling in the row 79 (ERCP) was not found; two different diagnostic modalities were mentioned - MRCP and ERCP
were mentioned
Ad. 5. We added images from the endoscopic procedure - a representative example of diagnosis and treatment of BS during ERCP
Ad 6. Future research direction is a head-to-head comparison of the results of biliary anastomoses that will differ in length of the donor's bile duct; it is mentioned at the end of our manuscript in the Conclusions section
Ad. 7 Conclusions are briefly mentioned at the end of the paper

Round 2

Reviewer 2 Report

Comments and Suggestions for Authors

Dear authors 

Thank you for the revisions that you have made to the manuscript. 

Just one final remark - the last paragraph, where you mention the conclusions, put it into a separate section, called Conclusions, in the same manner as Discussion.

Author Response

As it was mentioned in the Discussion section we analyzed the impact of CIT on BS development twice - in our previous study and currently. We obtained almost  the same results. We agree that prolonged CIT has an indisputable impact on graft ischemia, but CIT in our patients was relatively short and very rarely exceeded 8 hours in both study and control groups. Therefore it is difficult to incriminate CIT as the reason of anastomotic biliary strictures in our series.